# Geometry and UV-Vis Spectra of Au^3+^ Complexes with Hydrazones Derived from Pyridoxal 5′-Phosphate: A DFT Study

**DOI:** 10.3390/ijms24098412

**Published:** 2023-05-07

**Authors:** Oleg A. Pimenov, Konstantin V. Grazhdan, Maksim N. Zavalishin, George A. Gamov

**Affiliations:** General Chemical Technology Department, Ivanovo State University of Chemistry and Technology, Sheremetevskii pr. 7, 153000 Ivanovo, Russia; pimenov@isuct.ru (O.A.P.); grazhdan_kv@isuct.ru (K.V.G.); zavalishin00@gmail.com (M.N.Z.)

**Keywords:** gold(III), hydrazone, pyridoxal 5′-phosphate, DFT, molecular orbitals, UV-Vis spectra

## Abstract

Gold(III) complexes with different ligands can provide researchers with a measure against pathogenic microorganisms with antibiotic resistance. We reported in our previous paper that the UV-Vis spectra of different protonated species of complexes formed by gold(III) and five hydrazones derived from pyridoxal 5′-phosphate are similar to each other and to the spectra of free protonated hydrazones. The present paper focuses on the reasons of the noted similarity in electron absorption spectra. The geometry of different protonated species of complexes of gold(III) and hydrazones (15 structures in total) was optimized using the density functional theory (DFT). The coordination polyhedron of gold(III) bond critical points were further studied to identify the symmetry of the gold coordination sphere and the type of interactions that hold the complex together. The UV-Vis spectra were calculated using TD DFT methods. The molecular orbitals were analyzed to interpret the calculated spectra.

## 1. Introduction

The complexes formed by gold(III) are known to be isostructural and isoelectronic to platinum(II) compounds and can be seen as potential cytotoxic agents and antitumor drugs [1]. An interest in Au^3+^ complexes was revived in the mid-1990s [2,3,4] and has not extinguished yet, as recent papers show (see, e.g., reviews [5,6,7]). Despite their similarity with Pt^2+^ derivatives, it is the interaction with different proteins (in particular, thioredoxin reductase, which regulates cellular redox balance), not DNA, that seem to provide gold(III) complexes with biological activity [8,9,10,11]. However, Au^3+^ complexes are capable of damaging DNA *via* cleavage and oxidation (see, e.g., [12]), which can also be an important mechanism of biological action. Another advantage of Au use in medicine is its low toxicity for humans [13]. The possibility of using gold(III) complexes for the treatment of bacterial, fungal infections, and malaria is much less studied (a review [14] still accumulates most of the known data on gold(III) complexes as antibiotics; see also papers [15,16,17,18,19,20]). However, the most recent paper by Ratia et al. [21] shows the potential of gold(III) complexes in the most convincing manner.

The key disadvantages of gold(III) complexes is their instability in aqueous solutions, especially under physiological conditions. Neutral medium and biological reductant agents make gold(III) ions undergo either hydrolysis or reduction to Au^+^ or Au^0^ species, which changes their biological effect [22]. Moreover, the ligands evolving upon change of oxidation state of the metal ion can also show their own activity toward an organism [23]. The problem with metal complex stability can be solved if the appropriate ligands are chosen. Gold(III) binds efficiently with N,O-donor ligands since metal Lewis acidity corresponds best to nitrogen and oxygen Lewis basicity [24]. Therefore, molecules containing pyridine, bipyridine, and phenanthroline moieties, as well as macrocyclic ligands and porphyrins, are typical in gold(III) coordination chemistry [25,26,27,28,29].

The hydrazones derived from either pyridoxal or pyridoxal 5′-phosphate are tridentate ligands that form chelate complexes with d-metal ions in a stoichiometric ratio of metal:ligand = 1:2. The metal ion is tightly bound by imine nitrogen, the hydroxyl group in site 3 of the pyridoxal or pyridoxal 5′-phosphate residue, and carbonyl oxygen [30,31,32] (the phosphate group can contribute in the complexes of lanthanides(III) [33]). In addition, these hydrazones are low-toxic and membranotropic agents [34], which makes them especially attractive for developing new drugs based on hydrazones complexes (see, e.g., papers [35,36], where the antimicobacterial properties of complexes based on the hydrazones analogous to those derived from pyridoxal or pyridoxal 5′-phosphate are studied).

Our previous contribution reports on the stability of different protonated complexes of gold(III) with five hydrazones derived from pyridoxal 5′-phosphate [37]. These complexes were found to be relatively stable and formed with a high yield under physiological conditions. The authors of [37] also set several questions. One of them is why the electron absorption spectra of different protonated complexes are that similar to each other and those of free ligands. In the present contribution, we aim to answer it. To do so, it is necessary (1) to obtain the geometrical parameters of gold(III) complexes with the same hydrazones derived from pyridoxal 5′-phosphate (Figure 1); (2) to describe structural peculiarities of the coordination compounds; and (3) to compute UV-Vis spectra of gold(III) complexes and to compare them with those given in paper [37]. This goal is achieved using the density functional theory (DFT) calculations.

A brief review of the available literature data on DFT calculations of Au^3+^ complexes shows that gold(III) is prone to having square planar coordination, while the electron structure of the complexes is diverse [1,38,39,40,41,42,43]. Metal ions can contribute to LUMO [38,39,40], both HOMO and LUMO [1,41], or neither [42,43]. Ligand to metal charge transfer was reported in the paper [40]. The counter-ions complementing the coordination sphere of cation up to the square also often participate in frontier molecular orbital formation. This means that gold(III) complexes can show different UV-Vis spectra, even in complexes with the ligands possessing the same donor atoms (nitrogen and oxygen).

## 2. Results and Discussion

### 2.1. The Search of Probable Molecular Models

The search for probable molecular models of gold(III) complexes with the hydrazones derived from pyridoxal 5′-phosphate is based on our previous experience with a Zn(II) complex with pyridoxal 5′-phosphate 2-methyl-3-furoylhydrazone [44]. To obtain a demonstrative picture of complexation mechanism, we used the **PLP-F3H** hydrazone with a charge of –3*e* and analyzed the electronic density distribution for the most preferable geometry for cation coordination. Figure 2 demonstrates the molecular graph and molecular electrostatic potential (MEP) of **[PLP-F3H]^3−^** hydrazone.

As can be seen from the MEP distribution, the phosphate group was energetically more preferable to attracting the metal cation. However, the **[PLP-F3H]^3−^** hydrazone possessed three donor sites (O…N…O) that formed the chelate cavity that was the most reliable area to pick up the metal cation and form the stable complex. Gold(III) has been reported to form the complexes with Schiff bases in a similar way [45]. On the basis of this assumption, all molecular models are considered chelate complexes of gold(III) with hydrazones derived from pyridoxal 5′-phosphate. It should be noted that the hydrazones derived from pyridoxal 5′-phosphate (Figure 1) are denoted for simplicity as L, both here and below. The calculations of geometrical parameters of gold(III) complexes were carried out for both the AuClL^−^ molecule and for monoprotonated AuClHL^0^ and *bis*-protonated AuClH_2_L^+^ ones. To take into account the usage of HAuCl_4_ as a gold precursor for the synthesis of gold(III) complexes, the calculated molecular models are represented in Figure 3 as the hydrazone chelate complex, where the chloride anion is extra-coordinated to the gold cation. 

### 2.2. The Geometry of Gold(III) Complexes

The Cartesian coordinates of *C*_1_ symmetry models of gold(III) complexes with the hydrazones derived from pyridoxal 5′-phosphate are presented in Appendix A. To characterize the geometry of obtained molecules and study the influence of peripheral substitutions to coordination cavity, all considered molecular models were tested using SHAPE [46,47,48] software (version 7.4) to find the closest ideal reference polygons or polyhedra describing the AuO_2_NCl fragment. As a quantitative criterion, SHAPE calculated the continuous shape measure (CShM). According to the test, the best fit was obtained for the D_4h_ square planar polygon. In the series of AuClL^−^→AuClHL^0^→AuClH_2_L^+^, the value of CShM was decreasing. This result was in good accordance with the well-established square planar geometry of gold(III) complexes (see e.g., [1,38,39,40,41,42,43,49]). The *bis*-protonated AuClH_2_L^+^ species in all cases demonstrated the minimum distortion of the AuO_2_NCl fragment from the ideal square planar configuration. The serial protonation of AuClL^−^ species led to electronic density rearrangement that made the AuO_2_NCl fragment more symmetrical. The planar tetragon inscribed in the AuO_2_NCl fragment is presented in Figure 4 for the structure of the *bis*-protonatedfuroyl-2-hydrazide gold(III) complex. It possessed the least CShM = 0.630 (the closer to zero, the better this parameter can take the value of several units or even tens). 

The selected equilibrium distances of the AuO_2_NCl fragment of the considered gold(III) hydrazone complexes are provided in Table 1. 

According to DFT calculations, the geometry of the AuO_2_NCl coordination tetragon changed slightly along the series from **AuCl(PLP-F3H)** to **AuCl(PLP-INH)** (see Table 1 from the left to the right). The most significant change was observed for the Au–O2 bond distance as it was located closely to the position of the substituted moiety, and, thus, was affected the most. From **AuCl(PLP-T3H)** to **AuCL(PLP-INH)** bis-protonated forms, the Au–O2 bond distance increased by 0.009 Å. As a result, the coordination cavity size O1…O2 was enlarged by 0.006 Å, while the rest of the changes in bond distances did not exceed 0.004 Å. The most drastic alteration of the AuO_2_NCl structural parameters took place in series of AuClL^−^→AuClHL^0^→AuClH_2_L^+^, *viz.*, the systematical shortening of the Au–Cl distance was about 0.04 Å, while the elongation of the Au–N bond was more than 0.01 Å. It is interesting to note that the single protonation of hydrazone ligand L AuClL^−^→AuClHL^0^ was accompanied by the contraction of the coordination cavity size and the decreasing of O1…O2 by 0.009 Å; however, the next protonation step AuClHL^0^→AuClH_2_L^+^ led to the expanding of the cavity by more than 0.01 Å (see Table 1). All changes of the geometrical parameters were the consequence of the electronic density redistribution. To study the effect in detail, the analysis of the electron density distribution function *ρ(r)* is provided. 

### 2.3. The Electron Density Distribution Analysis

As mentioned above, the influence of ligand substituted fragments on the coordination tetragon of AuO_2_NCl in gold(III) hydrazone complexes is insignificant compared to the protonation effect within a single molecule. Therefore, as an example, the series AuClL^−^→AuClHL^0^→AuClH_2_L^+^ with L = **PLP-F3H** was selected to explore the electron density distribution peculiarities and, as a consequence, the nature of chemical bonds between gold(III) and hydrazones derived from pyridoxal 5′-phosphate. The main topological characteristics of electron density distribution function *ρ(r)* in the AuO_2_NCl fragment of **AuCl(PLP-F3H)** complex are provided in Table 2.

To visualize the topological peculiarities of *ρ(r)*, the molecular graph and Laplacian map in the plane of coordination cavity ONO are shown in Figure 5 for the *bis*-protonated **AuCl(PLP-F3H)** complex.

The Laplacian map (Figure 5) displays regions of local charge depletion (∇^2^*ρ*_b_ > 0) and concentration (∇^2^*ρ*_b_ < 0) for the **AuCl(PLP-F3H)** complex. The strongest charge depletion took place between the gold, chloride, and **PLP-F3H** ligand, which is also supported by a relatively small positive value of Laplacian ∇^2^*ρ*_b_ in the bond critical points (BCP) of the AuO_2_NCl fragment. Additionally, the ratio value |*λ_1_*/*λ_3_*| < 1 means the electron density was shifted from Au towards N, Cl, and O atoms (see Table 2). Despite this shifting, the relatively high electron density *ρ*_b_ characterizing the Au–N bond and negative value of total electronic energy *H*_b_ in BCP corresponded to an intermediate type of interatomic interaction (according to the terminology used in QTAIM by Bader). Such an interaction is typical for coordination and covalent polar bonds and has been studied in particular for the AuCl∙PPh_3_ complex in the crystal phase [50]. The slight deformation of local charge along the bond paths from N, Cl, and O atoms to Au corresponds to electron lone pairs of these atoms directed to the gold atom. 

The redistribution of the electron density in the AuO_2_NCl fragment can be evaluated from the changes in atomic charges. The atomic charges of the AuO_2_NCl fragment in the series of AuClL^−^→AuClHL^0^→AuClH_2_L^+^ are provided in Table 3 for the **AuCl(PLP-F3H)** complex.

The sequential protonation led to the decrease in the electron density in the AuO_2_NCl fragment. In particular, it can be seen from the increase in the positive charge on gold and the decrease in the negative charge on chlorine (Table 3). It is noteworthy that the Coulombic attraction between Au and Cl atoms (which could be estimated as *q*(Au) ∙ *q*(Cl)/*r_e_*^2^) remained almost unaltered (slightly decreasing) in the series of AuClL^−^→AuClHL^0^→AuClH_2_L^+^; however, the Au–Cl distance was noticeably shortened. This shortening can be explained by the reduction of electrostatic repulsion between negatively charged oxygen atoms O1 and O2, which was caused by the decrease in the chlorine negative charge and the total negative charge of O1 and O2. It allowed the chlorine ion to come closer to Au (Table 1). On the other hand, the observed elongation of the Au–N distance accompanying the shrinkage of the Au-Cl bond cannot be interpreted simply as the electrostatic interactions only. If this were true, the Au-N bond would become shorter; however, it was elongated (Table 1). The changes in the angle *∠*O1–Au–O2 (see the atom numbering in Figure 4) were negligible during the protonation steps, which meant that the Au–N elongation was caused by nitrogen shifting from Au. The protons effectively pulled the electron density, which led to the changes in the covalent bond lengths in the hydrazone ligand and, in particular, in the position of the N atom involved in the Au–N bond.

### 2.4. The UV–Vis Absorption Spectra Calculations

To determine the chemical validity of the proposed gold(III) hydrazone complexes, the theoretical UV–Vis absorption spectra were calculated and compared to the experimental ones. The energies of vertical electronic transitions and oscillator strengths were calculated by the TD DFT method CAM-B3LYP for all considered models and can be found in the Appendix A. To simulate the shape of the experimental UV–Vis absorption spectra in range 225–500 nm, the individual bands were described by Lorentz curves with a half-width of 47 nm. As a result, in Figure 6, the observed and simulated UV–Vis absorption spectra were presented for the **AuCl(PLP-F3H)** complex. For other models, the observed and the simulated UV–Vis absorption spectra are provided in the Appendix A (Appendix A). It can be seen from the data in Figure 6 that the positions of the absorption band of the simulated spectra were very close to the observed absorption maxima in the vicinity of 300 and 350 nm. In the series AuClL^−^→AuClHL^0^→AuClH_2_L^+^, the best fitting in all cases was obtained for AuClL^−^ and *bis*-protonated AuClH_2_L^+^ forms. The results of TDDFT calculations allow for an assignment of the experimental absorption bands. Table 4 summarizes the data on the vertical electronic transitions and oscillator strengths corresponding to the absorption band about 300 and 350 nm for all discussed complexes of gold(III) with hydrazones. 

It should be noted that the vertical electronic transitions were of complex composition. The effect of substitution led to weak changes of the electronic structure of the ligand, as is shown in Section 2.2. The same tendency was observed according to TD DFT spectra along the series from **AuCl(PLP-F3H)** to **AuCl(PLP-INH)** (see Table 4 from up to down) as the vertical electronic transitions energies (**λ_cal_**) varied within the narrow range. On the other hand, the qualitative changes in spectra accompanied the protonation process AuClL^−^→AuClHL^0^→AuClH_2_L^+^ according to our results. First, AuClL^−^→AuClHL^0^ led to the alteration of HOMO and LUMO composition. The shapes of the selected molecular orbitals (MO) for the considered complexes of gold(III) with hydrazones derived from pyridoxal 5′-phosphate are provided in Appendix A. The AuClL^−^ species possessed HOMO that fully consisted of *p* atomic orbitals of the ligand without any contribution from the phosphate group or gold ions, while LUMO was a combination of gold *d-*orbitals and *p*-orbitals of N, Cl, and O atoms (referred to as *d*Au-*p*), forming the coordination cavity. As a result, there was no HOMO→LUMO transition for AuClL^−1^ because such a transition has low probability. The first protonation step AuClL^−^→AuClHL^0^ led to the energy shift in MO. In particular, LUMO *d*Au-*p* became LUMO + 1, and following this, a HOMO→LUMO transition was observed in the spectra as a π→π * excitation into a S_2_ electronic state (see Table 4). The second protonation step resulted in the swapping of LUMO and LUMO + 1, which made a HOMO→LUMO transition for *bis*-protonated AuClH_2_L^+1^ species unlikely again. To make the discussion above more clear, the HOMO/LUMO energy level diagrams of gold(III) hydrazone complexes are presented in Appendix A. 

According to the MO composition for models AuClL^−^ and AuClHL^0^, the electronic transitions in the vicinity of 300 and 350 nm are indicative of a π→π * character without metal-to-ligand charge transfer. In the case of *bis*-protonated AuClH_2_L^+^ species, the transition into S_4_ electronic state (Table 4) occurred due to the electronic excitation, which involves the LUMO *d*Au-*p* and occupied MO consisting of gold *d-*orbital and *p*-orbitals of Cl atoms (referred to as *d*Au-Cl; see Appendix A). Therefore, according to our modeling, the electronic transition in the vicinity of 300 nm in *bis*-protonated AuClH_2_L^+^ species could not be assigned to the clear π→π * transition. 

According to the data (Table 4), it is possible to estimate the quality of the agreement between the simulated spectra (λ_cal_) and observed absorption maxima (λ_exp max_) in the vicinity of 300 and 350 nm. Appendix A presents a diagram of absolute values of the deviations |λ_cal_ − λ_exp max_| ascending for all those reviewed in the excited states shown in Table 4.

As can be seen from the diagram (Appendix A), the majority (21 cases) of deviations were less than 20 nm (which correspond to the relative error less than 5%). The minority (four cases) exceeded 35 nm (relative error was higher than 10%), wherein these four cases corresponded to the S_2_ excited states of monoprotonated AuClHL^0^ species (Table 4). In other words, the AuClHL^0^ absorption spectra were calculated with the least precision. However, the maximum value of deviation was 45.63 nm (corresponds to the relative error of 13.2%). Thus, in general, the TD DFT calculated spectra and observed spectral data were in satisfactory agreement.

The data in Table 4 show that the calculated UV-Vis spectra of different protonated species of the gold(III) complex (AuClL^−^, AuClHL^0^, and AuClH_2_L^+^, where L remains constant) are similar to each other. It also holds for the comparison between monoanions, neutral species, or monocations containing different hydrazones. The most probable reason for this similarity is the dominant contribution of ligand molecular orbitals into the MO of the complex, where electron transitions take place.

## 3. Materials and Methods

### Calculation Details

The calculations of probable molecular models of gold(III) complexes with the hydrazones derived from pyridoxal 5′-phosphate were carried out for the singlet electronic state using the Gaussian 09W program package [51]. The equilibrium geometrical parameters and normal mode frequencies (Appendix A) were calculated using the hybrid DFT computational method B3LYP [52]. The energies of vertical electronic transitions and oscillator strengths (UV-Vis spectra; see Appendix A) were calculated using the TDDFT [53] method CAM-B3LYP [54]. The two-component relativistic effective core potential (ECP60MDF) [55] was applied for the inner electronic shells of Au (1*s*^2^2*s*^2^2*p*^6^3*s*^2^3*p*^6^3*d*^10^4*s*^2^4*p*^6^4*d*^10^4*f*^14^). The valence shells (5*s*^2^5*p*^6^5*d*^10^6*s*^1^) were described by the (41s37p25d2f1g/5s5p4d2f1g) basis set cc-pVTZ-PP [56]. The H, C, N, O, P, and Cl atomic electronic shells were described by the all-electron cc-pVTZ basis set [57]. To take into account the solvent effect of water, all calculations were performed using the polarizable continuum model (PCM) [58]. The visualization of ball-and-stick models and molecular orbitals was performed using the *ChemCraft *program [59]. The topological analysis of electron density distribution function *ρ(r)* in the terms of Bader’s Quantum Theory Atoms in Molecules (QTAIM) [60] was carried out using the AIMAll Professional software (version 19.10.12) [61]. 

## 4. Conclusions

The different protonated species of gold(III) complexes with five hydrazones derived from pyridoxal 5’-phosphate were studied using density functional theory methods. The complexes studied included completely deprotonated hydrazone molecules, as well as monoprotonated and *bis*-protonated ligands. The geometry of the complexes was optimized at the DFT/B3LYP level, and the gold coordination sphere was found to be closest to a square planar geometry. The analysis of bond critical points revealed that the gold and N and O atoms, forming the chelating cavity of hydrazones, were bound due to interatomic interactions that were intermediate between ionic and covalent. The main result of this study confirmed that the calculated UV-Vis spectra of different protonated species of gold(III) complexes were similar to each other, as reported previously for experimental electron absorption spectra. The dominant contribution of ligand molecular orbitals into the MO of the complex, where electron transitions take place, is likely the reason for this similarity. We also observed an interesting change in the calculated UV-Vis spectra caused by the first step of protonation, which enables electron transition between frontier MOs, whereas in monoanionic or monocationic complexes, the HOMO→LUMO transition is hindered.

## Figures and Tables

**Figure 1 ijms-24-08412-f001:**
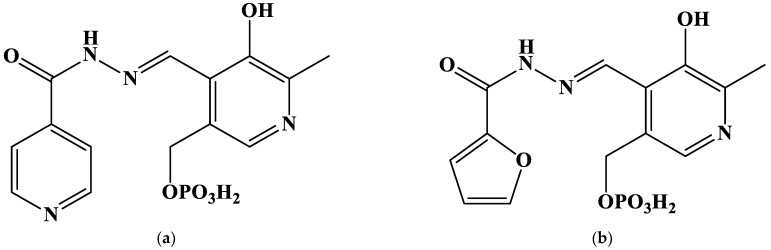
Structures of the studied hydrazones derived from pyridoxal 5′-phosphate and (**a**) isoniazid (**PLP-INH**); (**b**) furoyl-2-hydrazide (**PLP-F2H**); (**c**) thiophene-2-carbohydrazide (**PLP-T2H**); (**d**) 2-methylfuroyl-3-hydrazide (**PLP-F3H**); (**e**) thiophene-3-carbohydrazide (**PLP-T3H**).

**Figure 2 ijms-24-08412-f002:**
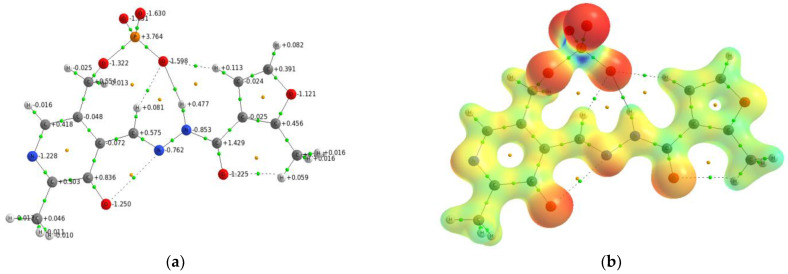
(**a**) The molecular graph with atomic charges for **[PLP-F3H]^3−^** hydrazone. The bond critical points are green, and the ring critical points are yellow. (**b**) The molecular electrostatic potential (MEP) is mapped on the isodensity surface (0.08 a.u.) for **[PLP-F3H]^3−^** hydrazone within the range of −0.459 a.u. (red) to +0.824 a.u. (blue).

**Figure 3 ijms-24-08412-f003:**
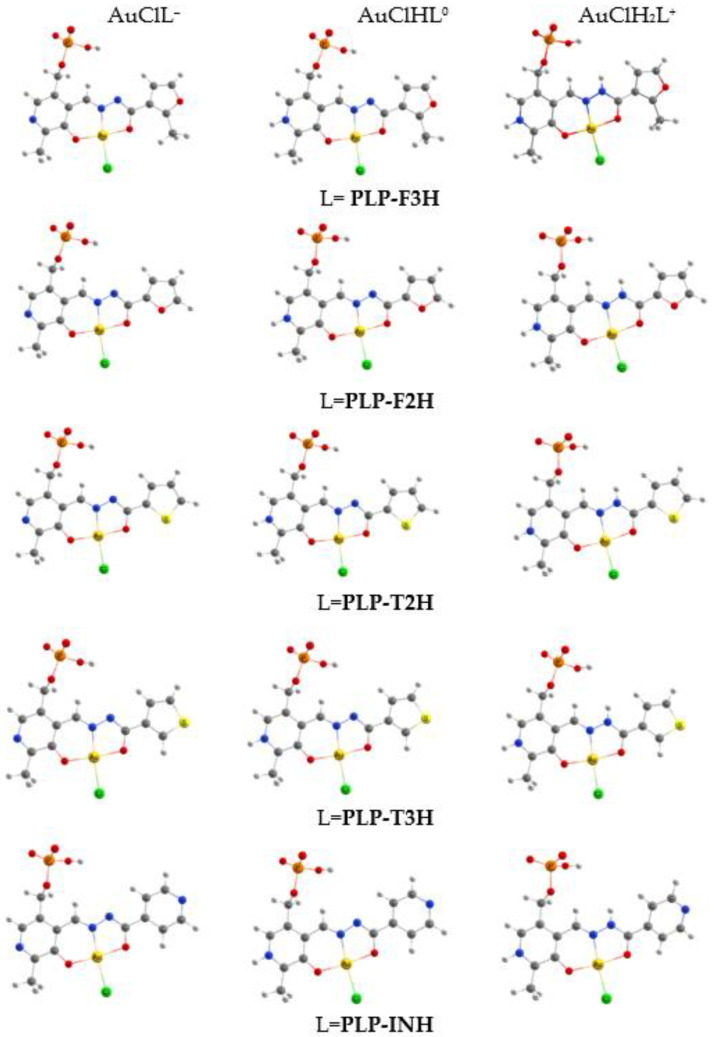
The *C*_1_ symmetry molecular models of gold(III) complexes with the hydrazones derived from pyridoxal 5′-phosphate.

**Figure 4 ijms-24-08412-f004:**
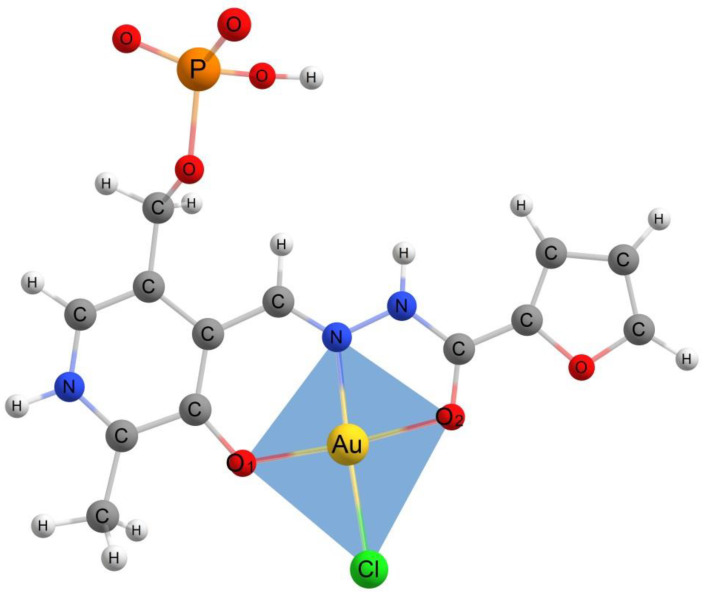
The inscribed in AuO_2_NCl fragment planar tetragon of the *bis*-protonatedfuroyl-2-hydrazide gold(III) complex and oxygen atoms numbered.

**Figure 5 ijms-24-08412-f005:**
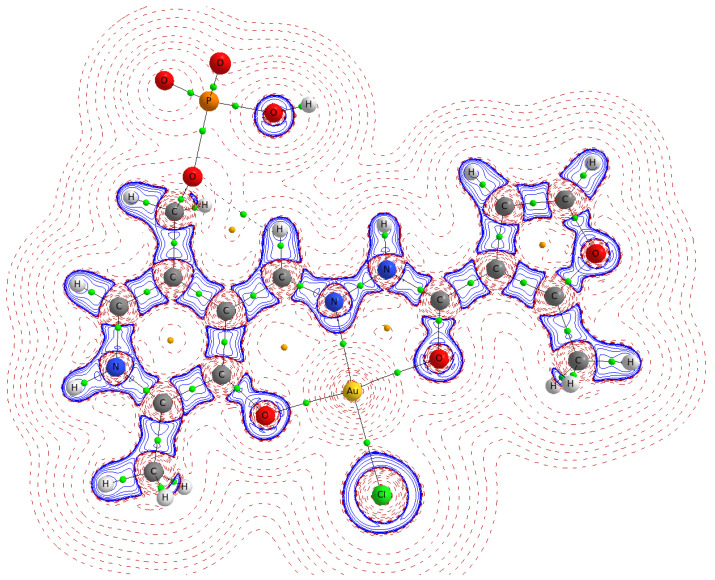
Molecular graph and Laplacian map in the plane of coordination cavity ONO for the *bis*-protonated **AuCl(PLP-F3H)** complex. The solid lines (blue) correspond to local charge accumulation (∇^2^*ρ_b_* < 0), while the dashed lines (red) represent a local charge depletion (∇^2^*ρ_b_* > 0). The bond critical points are green, and ring critical points are yellow.

**Figure 6 ijms-24-08412-f006:**
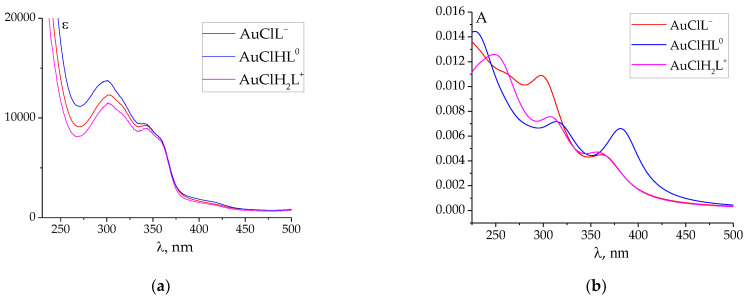
(**a**) The observed UV–Vis absorption spectra for the protonated species of the **AuCl(PLP-F3H)** complex (taken from [37]). (**b**) The simulated UV–Vis absorption spectra for protonated species of the **AuCl(PLP-F3H)** complex in the range of 225–500 nm. The theoretical individual bands were described by Lorentz curves with a half-width of 47 nm.

**Table 1 ijms-24-08412-t001:** Equilibrium distances (Å) of AuO_2_NCl fragment into gold(III) hydrazone complexes obtained by DFT calculations.

Distance	AuCl(PLP-F3H)	AuCl(PLP-F2H)	AuCl(PLP-T3H)	AuCl(PLP-T2H)	AuCl(PLP-INH)
*r*(Au–Cl)	2.334 *	2.332	2.332	2.333	2.331
	2.324	2.322	2.322	2.323	2.320
	2.294	2.293	2.292	2.294	2.292
*r*(Au–N)	1.987	1.989	1.987	1.989	1.987
	1.993	1.994	1.992	1.995	1.992
	2.000	2.000	1.999	2.000	2.001
*r*(Au–O1)	1.981	1.978	1.981	1.979	1.979
	1.983	1.981	1.984	1.981	1.982
	1.971	1.970	1.971	1.971	1.969
*r*(Au–O2)	2.003	2.006	2.003	2.005	2.005
	1.992	1.995	1.992	1.993	1.994
	2.015	2.017	2.014	2.016	2.023
*r*(O1…O2)	3.980	3.981	3.980	3.980	3.981
	3.971	3.972	3.972	3.971	3.972
	3.983	3.983	3.982	3.983	3.988

* The three rows up to down for each distance corresponding to AuClL^−^, AuClHL^0^, and AuClH_2_L^+^ forms.

**Table 2 ijms-24-08412-t002:** Topological parameters of *ρ(r)* in bond critical points of the AuO_2_NCl fragment into the **AuCl(PLP-F3H)** complex.

Interaction	*r_e_*	*ρ* _b_	∇^2^*ρ*_b_	*λ_1_*	*λ_2_*	*λ_3_*	*ε*	*δ*	*G* _b_	*V* _b_	*H* _b_
Au–Cl	2.334 *	0.662	+4.123	−0.097	−0.094	+0.363	0.032	0.885	+0.0761	−0.1095	−0.0334
	2.324	0.680	+4.036	−0.102	−0.098	+0.368	0.031	0.906	+0.0770	−0.1122	−0.0352
	2.294	0.732	+3.773	−0.113	−0.111	+0.381	0.021	0.977	+0.0796	−0.1201	−0.0405
Au–N	1.987	1.007	+7.919	−0.199	−0.190	+0.717	0.047	0.828	+0.1506	−0.2195	−0.0689
	1.993	0.995	+7.982	−0.194	−0.187	+0.712	0.042	0.814	+0.1499	−0.2174	−0.0675
	2.000	0.957	+8.939	−0.178	−0.174	+0.723	0.026	0.769	+0.1554	−0.2185	−0.0631
Au–O1	1.981	0.917	+10.112	−0.179	−0.178	+0.777	0.003	0.819	+0.1609	−0.2174	−0.0565
	1.983	0.903	+10.467	−0.174	−0.173	+0.781	0.007	0.793	+0.1626	−0.2171	−0.0545
	1.971	0.937	+10.236	−0.185	−0.184	+0.794	0.006	0.823	+0.1644	−0.2231	−0.0587
Au–O2	2.003	0.879	+9.761	−0.172	−0.168	+0.745	0.027	0.767	+0.1534	−0.2060	−0.0526
	1.992	0.903	+9.892	−0.179	−0.175	+0.764	0.023	0.784	+0.1577	−0.2132	−0.0555
	2.015	0.843	+9.794	−0.163	−0.160	+0.729	0.019	0.725	+0.1496	−0.1979	−0.0483

* The three rows up to down for each distance corresponding to AuClL^−^, AuClHL^0^, and AuClH_2_L^+^ forms; *r_e_* is the equilibrium distance (Å); *ρ*_b_ is the electron density (eÅ^–3^); ∇^2^*ρ*_b_ is the Laplacian (eÅ^–5^); *λ_1_*, *λ_2_*, and *λ_3_* are electron density Hessian matrix eigenvalues (a.u.); *ε* is the bond ellipticity; *δ* is the electron delocalization index; *G*_b_ is the kinetic energy density (a.u.); *V*_b_ is the potential energy density (a.u.); *H*_b_ is the total electronic energy density (a.u.) as a sum of *G*_b_ and *V*_b_.

**Table 3 ijms-24-08412-t003:** The atomic charges (*q*) according to the QTAIM schemes of the AuO_2_NCl fragment in the **AuCl(PLP-F3H)** complex.

Protonated Species	*q*(Au)	*q*(Cl)	*q(*N)	*q*(O1)	*q*(O2)
AuClL^−^	+1.137	−0.499	−0.751	−1.024	−1.024
AuClHL^0^	+1.165	−0.472	−0.729	−1.023	−1.010
AuClH_2_L^+^	+1.219	−0.396	−0.725	−0.991	−1.012

**Table 4 ijms-24-08412-t004:** Selected vertical electronic transitions (UV–Vis absorption spectra) calculated using the TDDFT/CAM-B3LYP method for gold(III) complexes with hydrazones derived from pyridoxal 5′-phosphate in aqueous solution.

Complex	Protonated Form	Excited State	λ_cal_ (nm)	λ_exp max_ (nm)	Oscillator Strength (*f*)	Composition *	Character
AuCl(PLP-F3H)	AuL^−^	S_3_	365.64	340	0.2094	112 → 115 (48%), 98 → 114 (16%)	π→π *
		S_6_	301.07	301	0.538	112 → 115 (59%), 111 → 115 (30%)	π→π *
	AuHL^0^	S_2_	382.48	341	0.4084	113 → 114 (61%), 112 → 114 (34%)	π→π *
		S_6_	315.34	301	0.2776	112 → 114 (36%), 109 → 114 (29%)	π→π *
	AuH_2_L^+^	S_4_	359.36	343	0.178	105 → 114(50%), 111 → 115 (26%)	*d*Au-Cl → *d*Au-*p*
		S_6_	310.78	302	0.3531	113 → 115 (45%), 111 → 115 (35%)	π→π *
Au(PLP-F2H)	AuL^−^	S_3_	371.03	345	0.3073	109 → 111 (87%)	π→π *
		S_6_	308.57	320	0.5777	108 → 111 (82%)	π→π *
	AuHL^0^	S_2_	390.63	345	0.5444	109 → 110 (88%), 107 → 110 (12%)	π→π *
		S_6_	316.81	319	0.3263	107 → 110 (78%), 104 → 110 (11%)	π→π *
	AuH_2_L^+^	S_4_	362.11	347	0.2398	101 → 110(50%), 108 → 111 (40%)	*d*Au-Cl → *d*Au-*p*
		S_6_	305.77	321	0.7109	106 → 111 (64%), 108 → 112 (13%)	π→π *
Au(PLP-T3H)	AuL^−^	S_3_	366.18	344	0.2188	113 → 115 (94%)	π→π *
		S_6_	300.15	305	0.5659	112 → 115 (84%)	π→π *
	AuHL^0^	S_2_	378.94	343	0.4116	113 → 114 (80%), 110 → 114 (10%)	π→π *
		S_6_	309.81	306	0.4295	110 → 114 (76%), 107 → 114 (13%)	π→π *
	AuH_2_L^+^	S_4_	359.5	344	0.1836	104 → 114 (49%), 112 → 115 (39%)	*d*Au-Cl → *d*Au-*p*
		S_7_	298.89	306	0.3441	109 → 115 (32%), 111 → 115 (20%)	π→π *
Au(PLP-T2H)	AuL^−^	S_3_	370.67	358	0.2977	113 → 115 (92%)	π→π *
		S_6_	308.23	312	0.5955	112 → 115 (86%)	π→π *
	AuHL^0^	S_2_	388.13	346	0.5268	113 → 114 (89%)	π→π *
		S_6_	315.74	308	0.3514	110 → 114 (65%), 111 → 114 (14%)	π→π *
	AuH_2_L^+^	S_4_	361.33	349	0.2318	104 → 114 (53%), 112 → 115 (42%)	*d*Au-Cl → *d*Au-*p*
		S_6_	308.2	320	0.6769	110 → 115 (56%), 112 → 116 (18%)	π→π *
Au(PLP-INH)	AuL^−^	S_3_	367.67	358	0.1659	112 → 114 (88%)	π→π *
		S_6_	298.85	305	0.4482	111 → 114 (90%), 103 → 113 (10%)	π→π *
	AuHL^0^	S_2_	371.42	347	0.2778	112 → 113 (93%)	π→π *
		S_6_	304.51	307	0.3935	108 → 113 (49%), 109 → 113 (18%)	π→π *
	AuH_2_L^+^	S_4_	361.58	346	0.1556	111 → 114 (56%), 103 → 113 (23%)	*d*Au-Cl → *d*Au-*p*
		S_12_	280.3	307	0.1657	110 → 114 (37%), 108 → 114 (18%)	π→π *

* The composition includes the first two most significant (%) transitions. The transitions with contribution less than 10% were omitted.

## Data Availability

Data are contained within the article or Appendix A.

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
