# Peer review of "Geometry and UV-Vis Spectra of Au3+ Complexes with Hydrazones Derived from Pyridoxal 5′-Phosphate: A DFT Study"

_ijms, 2023, doi:10.3390/ijms24098412_

Round 1

Reviewer 1 Report

The author discusses the potential of using gold(III) complexes with different ligands as a useful tool for researchers in fighting against antibiotic-resistant microorganisms. The study builds upon prior research that found that various protonated species of gold(III) complexes with specific hydrazones had similar UV-Vis spectra. To explore the similarity of electron absorption spectra further, the authors optimize the geometry of 15 complexes and analyze the bond critical points and coordination polyhedron. They evaluate molecular orbitals to interpret the calculated spectra. While the manuscript provides fresh insights into the electronic structure and bonding of gold(III) complexes, it requires a few minor revisions before publication in this journal.

1.     In the abstract, the author briefly mentions the potential for gold(III) complexes with different ligands to serve as agents against antibiotic-resistant microorganisms. However, the details of how these complexes exhibit this potential are not elaborated on in depth. It would be helpful for the author to provide more detailed information on the mechanisms behind the interaction between the complexes and microorganisms.

2.     The author should include an orbital energy level diagram for all the gold (III) complexes studied to aid in understanding the bonding and electronic structure of these metal-ligand complexes. The diagram would provide valuable insights into the stability and properties of the complexes.

3.     The author has presented data on the oscillator strength of all the complexes; however, they have not provided an interpretation of the significance of the data concerning these complexes.

The manuscript's English quality is satisfactory, but it could be refined in some areas. While the language used is usually concise and understandable, there are instances of flawed phrasing, improper word usage, and grammatical errors that obstruct the clarity and precision of the text. If given more attention to additional editing and proofreading, the manuscript's coherence and overall quality could be enhanced.

Reviewer 2 Report

Review Report:

In the manuscript, Pimenov et al., report optimized geometries and theoretical spectra of different Au(III)-hydrazone derivatives. In each of the complexes, the metal ion mainly prefers a distorted square planar geometry involving hydrazone N and two O donors. A gradual change in computed structures upon protonation is discussed. They further compare the computed UV-Vis spectra of deprotonated, mono- and bis-protonated complexes with the respective experimental spectrum which were reported previously.

I find the work important in the context of Au(III) chemistry. However, the presentation could be improved. In addition, I find there should be some additional discussion/comments related to the reported structures. Therefore, I request the authors to address the following issues. Only then, I recommend publication of the present work. I have used the following abbreviations, P-page number and L-line number. My comments are as follows:

Specific Comments:

1. The authors should introduce the full form of DFT at least once in their manuscript.

2. P1-L25: I expect a few references related to Au(III) binding to proteins.

3. P2-Figure 1: It is hard to distinguish between two molecules, c and d. I suggest using a separate box to represent the five hydrazone derivatives.

4. P3-L97: The authors only computed those structures where the metal ion forms a chelate complex. However, the molecular electrostatic potential shows that phosphate group is more negatively charged and therefore should strongly attract Au(III). I wonder whether they tried to optimize such a structure where the metal binds to the phosphate group. If so, then what is the stability difference from the structures reported here.

5. P4-Figure 3: In case of mono- and bis-protonated species, I see there is only one site shown (for the bis- two protonation centers) where the additional proton binds. I wonder whether they considered any other protonation sites. If so, what’s the relative energy of those protonation sites?

6. P6-L148 and L-151: Please correct typos, more then → more than.

7. P6-L151: The authors only mention that protonation induced structural changes on the basis of electron density distribution. However, I find the need for a discussion in the manuscript at least for one complex as the structural changes are quite significant, in particular for Au-N.

8. P7-L206: The authors mention that the calculated and observed spectra closely agree, without any discussion on such agreement. I encourage some sentences in this regard.

9. P8-Figure 6: It is advised to indicate the y-axis.

10. P8-Figure 6: It is confusing to follow the legends in the experimental and computed spectra. I recommend the authors to be consistent with the colors and order. In addition, AuH2L_cam-b3lyp should be read as AuH2L_cam-b3lyp.

11. P8-L231: In case of AuHL, the computed relative intensity of the two transitions between ~300-400 nm is contrary to the experimental one. I recommend the authors to address this.

12. P10-L283: The authors mention that computed spectra of different protonated species are similar to each other, which is however not the case in particular for AuHL as seen from Figure 6. Therefore, I recommend revising this statement.

General Comments:

1. The English grammar could be improved (they used at Figure, which is not correct).  

2. I find some of the sentences are quite long and therefore hard to follow. The authors should consider this for better readability.

The English could be improved. 
